# Clinical Implications of Multi-Drug Resistant Organisms’ Gastrointestinal Colonization in an Internal Medicine Ward: The Pandora’s Box

**DOI:** 10.3390/jcm11102770

**Published:** 2022-05-14

**Authors:** Ombretta Para, Lorenzo Caruso, Eleonora Blasi, Caterina Pestelli, Giulia Pestelli, Stefano Guidi, Giacomo Fedi, Igor Giarretta, Fabrizio Maggi, Tiziana Ciarambino, Carlo Nozzoli, Francesco Dentali

**Affiliations:** 1Department of Emergency Medicine, Careggi University Hospital, 50134 Florence, Italy; crslorenzo@gmail.com (L.C.); eleonora.blasi@gmail.com (E.B.); caterinapestelli29@gmail.com (C.P.); giulia.pestelli@gmail.com (G.P.); steguidi92@gmail.com (S.G.); fedax93@gmail.com (G.F.); nozzolic@aou-careggi.toscana.it (C.N.); 2Department of Medicine and Surgery, Insubria University, 21100 Varese, Italy; igor.giarretta@asst-settelaghi.it (I.G.); fabrizio.maggi@asst-settelaghi.it (F.M.); francesco.dentali@asst-settelaghi.it (F.D.); 3Laboratory of Microbiology, Insubria University, 21100 Varese, Italy; 4Internal Emergency Department, Hospital of Marcianise, 80125 Caserta, Italy; tiziana.ciarambino@gmail.com; 5Department of Medicine, Insubria University, 21100 Varese, Italy

**Keywords:** sepsis, multi-drug resistant organism, rectal swab

## Abstract

**Background:** Multi-drug resistant organisms (MDRO) are an emerging health problem with an important impact on clinical outcome in Intensive Care Units (ICUs) and immunocompromised patients. Conversely, the role of MDRO colonization in Internal Medicine is less clear. The objective of our study is to evaluate the clinical impact (namely sepsis development, in-hospital and 30-days mortality, and re-hospitalization) of MDRO colonization in Internal Medicine. **Methods**: Patients admitted to our Internal Medicine Unit between January 2019 and March 2020 were potentially includible. Outcomes in patients with a positive rectal swab for MDRO (RS+) and in patients without a RS+ were compared. Results of the multivariate analyses were expressed as Odds Ratios (ORs) and the corresponding 95% Confidence Interval (CI). **Results**: In a cohort of 2147 patients, 77 patients with RS+ were consecutively identified; 377 patients with a rectal swab negative for MDRO were randomly selected from the same cohort (five for each patient with RS+). At the multivariate analysis, RS+ was associated with an increased risk of sepsis development during hospitalization (OR 4.18; 95% CI, 1.99–8.78) and with death or re-hospitalization at 30 days (OR 4.79; 95% CI, 2.79–8.23), whereas RS+ did not appear to be associated with death during hospitalization or need for ICU transfer. **Conclusions: Our results suggest for the first time a prognostic role for RS+ in Internal Medicine. Thus, assessment of rectal swab at hospital admission appears useful even in this setting.** However, larger prospective studies and a cost–benefit analysis are needed to confirm our preliminary findings.

## 1. Introduction

Antibiotic resistance is currently one of the greatest threats for public health, both in Europe and worldwide [1,2]. Recently, a comprehensive assessment of the global burden of antimicrobial resistance (AMR) estimated 4.95 million deaths associated with bacterial AMR in 2019, including 1.27 million deaths attributable to bacterial AMR. The latest surveys, based on data from *EARS-Net,* show that in UE/SEE, more than 670,000 infections per year are caused by resistant bacteria, with 33,000 consequent deaths [3]. In such contest, it is crucial to obtain data from microbiological surveillance in order to plan educational interventions and “*Antimicrobial Stewardship*” programs, aimed to prevent and control multi-drug-resistant infections. [4,5]. Prompt identification of multi-drug-resistant organism (MDRO) carriers is fundamental to control nosocomial infections, since functional isolation of multi-drug-resistant infected or colonized patients is one of the main precautions required to prevent in-hospital spread of these pathogens and also since MDRO are known to be potential sources of cross-transmission [6].

During the last few years, the burden of antibiotic resistance in the European Union (EU) and in the European Economic Areas (EEA) has been increasing for a number of bacteria. Among adults, such burden increases with age, suggesting that in the aging EU and EEA population, it may even be further enhanced [7]. Italy and Greece have the greatest burden of infections due to antibiotic-resistant bacteria, suggesting a more threatening situation in these two countries. Of note, it has been estimated that about 75% of the total burden of infections with antibiotic-resistant bacteria in the EU and the EEA countries is associated with health care, thus underlying the need for dedicated programs to control MDRO’s hospital colonization.

It is well known that in Intensive Care Units (ICU) and in high-risk patients (i.e., onco-hematological patients and in solid-organ transplant recipients), MDRO colonization is associated with an increased risk of infection and adverse outcomes [8,9,10].

On the other hand, only a few studies have assessed the role of gastrointestinal colonization with MDRO in Internal Medicine. The usefulness of a prompt identification of MDRO colonization in these extremely frail patients is also not clear.

In this study, we therefore aimed to identify the clinical impact of the presence of MDRO gastrointestinal colonization in patients admitted to an Internal Medicine Ward.

## 2. Materials and Methods

### 2.1. Population

Patients admitted to the tertiary care Internal Medicine Unit at Careggi University Hospital in Florence between 1 January 2019 and 15 March 2020 were potentially includible in the study. For the enrollment, we used an electronic digitized medical record (Archimed^®^ medical software version 6.20 by B. Dannaoui, Florence, Italy).

Patients with a rectal swab positive for MDRO were identified in this large cohort of patients. Subsequently, for each patient with rectal swab positive for MDRO, 5 patients with rectal swab negative for MDRO were randomly selected from the same cohort. All the patients had a rectal swab performed by a trained nurse on the day of admission to the Internal Medicine Unit. In our microbiology laboratory, rectal swabs were steaked on selective culture media, and the strains were identified using MALDI-ToF technology. According to our center surveillance program, based on the current Regional Law and Guidelines, the following bacteria were detected: Carbapenem-Resistant Enterobacteriaceae (CRE), Carbapenem-Resistant Acinetobacter baumanii (CRAB), Carbapenem- Resistant Pseudomonas aeruginosa (CRPsA) and Vancomycin-Resistant Enterococci (VRE). A rectal swab was considered positive for MDRO if at least one of these bacteria, CRE, CRAB, CRPsA or VRE, was present [11]. On every rectal swab that turned out to be positive for MDRO, the lab specified the mechanism of resistance through real-time PCR. The drug susceptibility test was not routinely performed for all the isolated Gram-negative rods. However, if the isolated organism had a remarkable clinical or epidemiological significance, further investigation was prompted and the antibiotic susceptibility was determined according to the EUCAST criteria [12]

In agreement with the 2016 *Consensus Conference* [13], sepsis was confirmed through the quick SOFA (*qSOFA) score,* which allows to quickly evaluate patients with suspected sepsis hospitalized in Emergency or Internal Medicine departments.

Since the primary endpoint of the study was the development of sepsis during hospitalization, patients affected by sepsis at the time of hospital admission were excluded.

For each patient, the following data at the time of hospital admission were collected:Demographic data;Main reason for hospital admission;Comorbidities (i.e., hypertension, diabetes, dementia, heart failure);Treatment at the time of hospital admission (with focus on corticosteroids and immunomodulatory agents);Current and previous (3 months) antibiotic use;Presence of indwelling urinary and/or vascular catheter;Long-term care facility stay (e.g., nursing home);Previous hospitalization or stay in health residence in the previous 12 months.

Information on clinical events during hospitalization and at 1-month follow-up was first collected using patients’ electronic digitized medical records. Furthermore, for the purpose of this study and in order to guarantee the most accurate information collection, we contacted all included patients with missing data at follow-up by means of a visit at the center or a telephone conversation.

At the time of evaluation, information on vital status and re-hospitalization within 30 days after the index hospitalization was recorded. For all reported events during follow-up, accurate evaluation of source documentation was requested. Only objectively diagnosed events were considered.

The primary endpoint of our study was to identify the clinical impact of rectal swab positive for MDRO, namely the development of sepsis during hospitalization.

The secondary endpoint was a composite of in-hospital mortality and transfer to Sub Intensive/ICUs.

The tertiary endpoint was to evaluate 30-day mortality or in-hospital readmission.

Finally, we assessed the independent role of rectal swab positive for MDRO and of other comorbidities in predicting in-hospital and 30-day mortality as individual endpoints.

The study was performed in accordance with the Declaration of Helsinki and local regulations. The protocol was approved by the Institutional Review Board or Ethics Committee of our center which waived the need for written informed consent due to the retrospective design of our study.

### 2.2. Statistical Analysis

The study was carried out and reported according to the STROBE guidelines for observational studies [14].

The normality of data distribution was assessed using the Shapiro–Wilk test. Continuous variables were expressed as mean plus or minus standard deviation (SD) or as median with interquartile range (IQR), as appropriate. Categorical data were reported as counts and percentages.

Categorical variables were compared using Chi-squared or Fisher’s test, as appropriate. Continuous variables were compared with Student’s test or Mann–Whitney U-test, when appropriate.

Every variable associated with an outcome of the study with a *p*-value < 0.10 (entry-level) was included in a multivariate binary logistic regression. Stepwise elimination was performed to finalize the independent predictors of the multivariate models.

Statistical significance was reached when the *p*-value was < 0.05 (two-tailed). Results of the multivariate analyses were expressed as Odds Ratios (ORs) and the corresponding 95% Confidence Interval (CI). Statistical analyses were performed using STATA-16/MP (StataCorp LP, College Station, TX, USA).

## 3. Results

Between January 2019 and March 2020, 2147 consecutive patients admitted to Internal Medicine I (38 beds) of Careggi University Hospital were screened for participation in this study; of those, 1632 were evaluated with a rectal swab (76.0%), mainly for patients’ refusal, and 103 resulted positive for MDRO (6.31%, 95% CI 5.20–7.62% of the evaluated patients). Of these potentially includible patients, 26 patients were not included for the following reasons: 14 patients were septic at the time of hospitalization, and 12 had been previously hospitalized in our unit during the study time. Thus, 77 patients with an MDRO positive rectal swab and 377 patients with an MDRO negative rectal swab were selected and included in the study.

Patient characteristics are summarized in Table 1. The mean age was 73.36 years (SD; 16.28) and 249 patients (70.0%) were male. Hypertension was the most common comorbidity with 209 patients affected (46.0%) followed by solid active neoplasm (23.3%) and heart failure (20.0%); 76 patients (16.7%) had diabetes; 70 patients (15.4%) had dementia, 65 patients (14.3%) had chronic kidney disease and 59 patients (13.0%) had chronic obstructive pulmonary disease. A history of cerebrovascular disease was reported by 56 patients (12.3%), while coronary artery disease was reported by 53 patients (11.7%). Furthermore, 41 patients (9.0%) had a BMI > 30 Kg/m^2^ and 21 (4.6%) were affected by hematological neoplasm. Almost half of the enrolled population (233 patients; 51.3%) had at least two comorbidities and 27.3% (126 patients) were under immunomodulatory or immunosuppressive therapy.

Most of the patients were admitted for pneumonia (21.4%), for acute complication of cancer (9.9%) or for heart failure (9.7%). For the study purpose and analysis, hospitalization diagnoses were divided into four categories: infectious diseases (27.1%), atherothrombotic diseases (10.1%), exacerbation of chronic diseases (36.1%) and other diseases (24.4%).

The mean length of hospitalization was 8.41 ± 7.66 days; 37 (8.1%) patients had to be transferred to ICUs while 30 (6.6%) subjects died during hospitalization.

VRE was the most frequently isolated microorganism (64%) followed by *Klebsiella pneumoniae* carbapenemase-producing (KPC 23%), *Escherichia coli* (2%), *Acinetobacter* spp. (1%) and Pseudomonas aeruginosa or other combinations of more than one MDRO (3%). Positivity for multiple MDRO was found in seven patients (Figure 1).

The prevalence of hypertension, heart failure, dementia, chronic kidney disease, chronic obstructive pulmonary disease, history of cerebrovascular events and hematological neoplasm was significantly higher in RS+ patients compared with RS- patients. In addition, the duration of hospitalization was significantly higher in patients with RS+ for MDRO compared with patients with RS negative for MDRO. RS was more frequently positive for MDRO in patients hospitalized for infectious diseases or for exacerbation of chronic diseases. Conversely, the number of patients with RS+ for MDRO was lower in subjects hospitalized for atherothrombotic disease or other diseases (Appendix A).

The primary outcome of sepsis during hospitalization was registered in 33 patients (7.27%; 95% IC, 4.88–9.66). Results are reported in Table 2.

At the univariate analysis, hospital admission for infectious diseases (42.4% vs. 25.9%; *p* < 0.05) and RS+ for MDRO (42.4% vs. 15.0%; *p* < 0.001) were significantly associated with an increased risk of developing sepsis during hospitalization whereas there was a non-statistically significant trend for the association between the presence of two or more comorbidities and development of sepsis.

At the multivariate analysis, RS+ for MDRO was the only variable significantly associated with development of sepsis during hospitalization (OR 4.18; 95% IC, 1.99–8.78).

The secondary outcome of death during hospitalization or need to be transferred to intensive care units was recorded in 67 patients (14.7%; 95% IC, 11.4–18.0). Results are reported in Table 3. None of the variables investigated was significantly or marginally associated with this endpoint at the univariate analysis. Therefore, the multivariate analysis was not performed.

In-hospital mortality was 6.6% (95% CI, 3.8–8.2). At the univariate analysis, age > 75 years and sepsis development during hospitalization were significantly associated with in-hospital mortality (80.0% vs. 51.9%; *p* < 0.01 and 16.7% vs. 6.6%; *p* < 0.05 respectively). There was a non-statistically significant trend for the association between the presence of two or more comorbidities and in-hospital mortality (66.7% vs. 50.2%; *p* = 0.08). On the other hand, RS + for MDRO was not associated with an increased risk of in-hospital mortality. At the multivariate analysis, age > 75 years, only, appeared to be associated (OR 3.28; 95% CI, 1.26–8.50) (Appendix A).

A 30-day follow-up was available in 433 patients (95.4% of the whole population), of which 75 (18.6%; 95% IC, 14.8–22.4) were re-hospitalized and 19 died (4.7%; 95% IC, 2.63–6.77) (5 patients died after being re-hospitalized). Including the 30 patients who died during hospitalization, the tertiary outcome thus occurred in 119 subjects (27.5%; 95% IC, 23.3–31.7). Results are reported in Table 4.

At the univariate analysis, hospitalization RS + for MDRO (32.8% vs. 9.24%; *p* < 0.001) and development of sepsis during hospitalization (10.9% vs. 5.1%; *p* < 0.05) were significantly more frequent in patients who died or were re-hospitalized at 30 days. There was a non-statistically significant trend for the association between this endpoint and hospitalization for infectious disease whereas hospitalization for other reasons was significantly less frequent in patients who died or were re-hospitalized at 30 days (18.5% vs. 29.9%, *p* <0.05).

At the multivariate analysis, RS+ for MDRO (OR 4.79; 95% IC, 2.79–8.23) was the only variable significantly associated with death or re-hospitalization at 30 days.

The 30-day mortality was 10.8% (95% CI, 7.94–13.6). At the univariate analysis, age > 75 years (71.4 vs. 51.6; *p* < 0.01), sepsis development during hospitalization (14.3% vs. 6.42%; *p* < 0.05) and the presence of two or more comorbidities (65.3% vs. 49.6%; *p* < 0.05) were significantly associated with 30-day mortality. On the other hand, RS + for MDRO was not associated with an increased risk of 30-day mortality. At the multivariate analysis, age > 75 years appeared only to be associated (OR 2.02; 95% CI, 1.01–4.00) (Appendix A).

## 4. Discussion

In our large cohort of 2147 consecutive patients admitted to Internal Medicine Units between January 2019 and March 2020, a rectal swab positive for MDRO was not uncommon, occurring in about 6.31% of all evaluated patients.

Furthermore, as seen in patients admitted to ICUs and in high-risk patients, such as onco-hematological patients or patients receiving a solid-organ transplant, MDRO colonization was associated with an increased risk of adverse outcome. In particular, MDRO colonization at admission was associated with an increased risk of developing sepsis during hospitalization and with an increased risk of death and hospital readmission at 30 days. Conversely, in our population, mortality or transfer to Sub Intensive/ICUs during the index hospitalization did not appear to be associated with rectal swab positive for MDRO.

Although quite variable in different studies, colonization with MDRO is moderately common, especially in rehabilitation, post-acute units and ICUs [15]. Prevalence of MDRO in our study appeared to be in line with previous studies performed in acute care departments, different from ICUs [15].

A number of studies have been conducted to investigate risk factors related to rectal colonization by MDRO. Several studies and meta-analyses, performed on patients in the intensive care unit settings, have clearly demonstrated that MDRO colonization is a marker for disease severity, and is associated with an increased risk of subsequent infection [16].

Similar results in non-ICUs settings were reported in patients receiving a transplant, in hematological and in cirrhotic patients [9,10].

In 2007, Reddy et al. [17] reported results from a large study (17,872 patients) assessing the prevalence of extended-spectrum beta-lactamase-producing Enterobacteriaceae (ESBLE) on rectal colonization in a large cohort of high-risk inpatients, including patients hospitalized in the medical ICU, in the surgical ICU, in the solid-organ transplant unit and in the hematology/oncology unit. The rate of ESBLE colonization doubled during the 6-year study period, increasing from 1.33% in 2000 to 3.21% in 2005. Of 413 patients colonized with ESBLE, 8.5% developed a subsequent ESBLE bloodstream infection, suggesting a potential role of MDRO in the development of severe infections.

In a multicenter prospective study on hematological inpatients [9], MDR rectal colonization occurred in 6.5% of patients (144 of 2226 patients) and predicted a 16% probability of MDRO-related bloodstream infections. The 3-month overall survival for the whole cohort was 88.36%, and was significantly lower in patients colonized with VRE (77.78%) and carbapenem-resistant Gram-negative bacteria (83.57%) in comparison to those colonized with ESBLE (96.8%).

Bert et al. [18] preoperatively screened for ESBLE fecal carriage i 710 transplant patients, revealing a prevalence of this multi-drug resistance in about 4.5% of patients. After a multivariable analysis considering several other potential risk factors, pretransplant ESBLE fecal carriage was significantly associated with the development of infection after surgery.

In a more recent small prospective study on patients who underwent liver transplantation, Massa et al. [10] confirmed results from the previous study by Bert et al. [18], and found that ICU length of stay was significantly longer and mortality was significantly higher in patients colonized with carbapenem-resistant Gram-negative bacteria compared to non-colonized patients.

Interestingly, we were not able to demonstrate any independent role of rectal swab positive for MDRO and of other comorbidities in predicting in-hospital and 30-day mortality obtained as individual endpoints. Age seems to be the only factor significantly increasing the risk of mortality in our study. In the study by Barrasa-Villar et al. [19], 1000 patients (324 with MDRO and 676 patients with susceptible strain) were included; said patients had to have a hospital acquired infection and they had to have been admitted to the hospital in any unit except for obstetrics’, pediatrics’ or psychiatry’s services. As specified above, we included 77 patients with RS+ for MDRO, the rectal swab had to be routinely performed and many patients in our study did not have an infection at the time of inclusion. Additionally, for study purposes, all the patients included had to be admitted to our Internal Medicine Unit. Thus, different results from the existing literature [17] may be due to the differences in the setting in which the studies were performed, as well as to the limited sample of patients with rectal swab positive for MDRO included in our study.

VRE and KPC are the two most frequent bacteria isolated on rectal swabs in our population, accounting for about 75% of the MDRO. In particular, according to the European Centre for Disease Prevention and Control (ECDC) report, the increase in the percentage of Vancomycin-resistant isolates of E. faecium in the EU, in the last few years, appears of particular concern [20]. Data on the antibimicrobial resistance in Italy substantially confirm our results, also pointing out the levels of CRE and CRAB. Therefore, antimicrobial resistance appears to be a major public health threat in our country. Hence, the introduction of appropriate measures to reduce unnecessary antibiotic use while improving infection control using systematic surveillance programs seems of paramount importance.

The results of our study may have some important implications for clinical practice. Considering the frequency of MDRO identified in our study and its association with sepsis as well as with mortality and hospital readmission at 30 days, assessment of rectal swab at hospital admission appears to be useful in Internal Medicine as in other clinical settings (e.g., ICUs, onco-hematological patients and patients receiving a solid-organ transplant). Although the extreme frailty of our patients appeared to justify the routine surveillance of MDRO in Internal Medicine, larger prospective studies and a cost–benefit analysis are needed to confirm our preliminary findings and to determine the usefulness of a surveillance program in Internal Medicine.

Our study has some limitations. First, the study was of retrospective and monocentric nature. Second, the analysis was limited to a single Internal Medicine Unit, in a single hospitalized setting, which returns an epidemiological picture that could be significantly different from other hospitals and should, thus, be generalized with extreme caution. Furthermore, since only five patients with a rectal swab negative for MDRO were randomly selected for each patient with a rectal swab positive for MDRO, we ended up with a relatively small number of patients included. Thus, due to this limited number of patients, we were unable to assess a number of other potentially important prognostic factors and we could not reliably exclude false positive and false negative results at our multivariate analyses. Last, we did not perform any analysis assessing potential risk factors for the development of MDRO since we believe this is outside the aim of our study. Thus, we could not make any inference on this important issue.

In conclusion, for the first time, the results of our study suggest a prognostic role of rectal swab positive for MDRO in patients admitted to Internal Medicine that show a three times greater risk of developing sepsis during hospitalization and a significantly higher risk of death or re-hospitalization at the 30-day follow-up. However, further large-scale prospective studies are needed to confirm our preliminary findings.

## Figures and Tables

**Figure 1 jcm-11-02770-f001:**
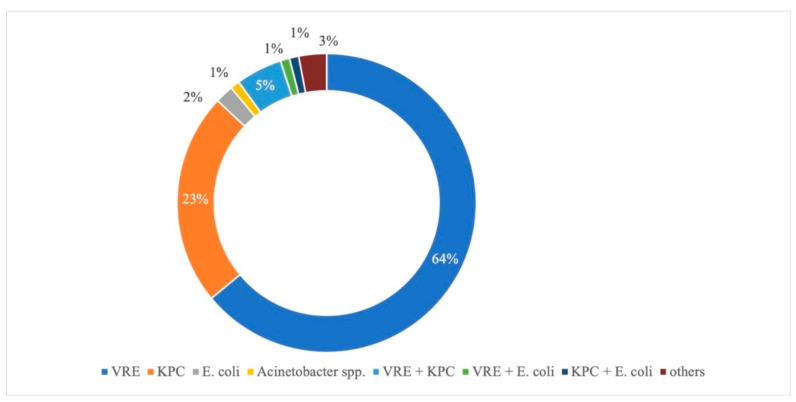
Multi-resistant microorganisms isolated by rectal swab. VRE: Vancomycin-Resistant Enterococcus; KPC: *Klebsiella pneumoniae* carbapenemase-producing.

**Table 1 jcm-11-02770-t001:** Epidemiological and clinical findings for the population included in the study and differences between patients with or without rectal swab positive for multi-drug resistant organism.

	General Population (454)	RS Positive (Tot *n* = 77)	RS Negative (Tot *n* = 377)	*p*
Demographic factors				
Age (years; mean ± SD)	73.36 ± 16.28	76.09 ± 15.31	72.81 ± 16.44	0.10
Male (*n*, %)	249 (54.8)	42 (54.5)	207 (55.0)	0.93
Immunomodulatory therapy (*n*, %)	124 (27.31)	22 (28.6)	102 (27.1)	0.78
Comorbidities				
Hypertension (*n*, %)	209 (46.0)	57 (74.0)	152 (40.3)	**<0.001**
Solid active neoplasm (*n*, %)	106 (23.3)	13 (16.9)	93 (25.2)	0.14
Heart failure (*n*, %)	91 (20.0)	29 (37.6)	62 (16.4)	**<0.001**
Diabetes (*n*, %)	76 (16.7)	20 (25.9)	56 (14.8)	0.17
Dementia (*n*, %)	70 (15.4)	31 (41.2)	39 (10.3)	**<0.001**
Chronic Kidney disease (*n*, %)	65 (14.3)	24 (31.2)	41 (10.9)	**<0.001**
Chronic obstructive pulmonary disease (*n*, %)	59 (13.0)	18 (23.3)	41 (10.9)	**<0.01**
History of Stroke/TIA (*n*, %)	56 (12.3)	20 (25.9)	36 (9.54)	**<0.001**
Coronary artery disease (*n*, %)	53 (11.7)	13 (16.8)	40 (10.6)	0.12
Obesity (*n*, %)	41 (9.0)	8 (12.9)	33 (8.7)	0.65
Hematological neoplasm (*n*, %)	21 (4.6)	8 (10.3)	13 (3.4)	**<0.01**
≥2 comorbidities (*n*, %)	233 (51.3)	53 (68.8)	180 (47.7)	**<0.05**

SD: standard deviation; RS: rectal swab; TIA: transient ischemic attack.

**Table 2 jcm-11-02770-t002:** Primary outcome, incidence of sepsis during hospitalization.

Variable	Sepsis (33)	Not-Sepsis (421)	Univariate Analysis	Multivariate Analysis	
			*p*	OR	CI
Male gender (*n*, %)	18 (56.2)	231 (54.9)	0.88		
Age (years; mean ± SD)	75.1 ± 18.3	73.2 ± 16.1	0.53		
Admission for ID (*n*, %)	14 (42.4)	109 (25.9)	**0.04**		
Admission for ATD (*n*, %)	2 (6.1)	44 (10.4)	0.42		
Admission for CDr (*n*, %)	8 (24.2)	156 (37.0)	0.14		
Admission for other reason (*n*, %)	9 (27.3)	111 (26.4)	0.91		
Solid active neoplasm (*n*, %)	8 (24.2)	98 (23.4)	0.91		
≥2 comorbidities (*n*, %)	22 (66.7)	211 (50.1)	**0.07**		
Immunomodulatory therapy (*n*, %)	8 (24.2)	116 (27.6)	0.68		
Rectal swab positive for MDRO (*n*, %)	14 (42.4)	63 (15.0)	**<0.001**	**4.18**	**1.99–8.78**

OR: odd ratio; CI: confidence interval; ID: infectious disease; ATD: atherothrombotic disease; CDr: exacerbation of chronic disease; MDRO: multi-drug resistant organism.

**Table 3 jcm-11-02770-t003:** Secondary outcome, incidence of death or need to be transferred to a higher intensity care unit during hospitalization.

Variable	Secondary Outcome (67)	Controls (387)	Univariate Analysis
			*p*
Male gender (*n*, %)	40 (59.7)	209 (54.1)	0.39
Age (years; mean ± SD)	74.1 ± 14.7	73.2 ± 16.5	0.67
Admission for ID (*n*, %)	13 (19.4)	110 (28.4)	0.12
Admission for ATD (*n*, %)	9 (13.4)	37 (9.6)	0.33
Admission for CDr (*n*, %)	27 (40.3)	137 (35.4)	0.44
Admission for other reason (*n*, %)	18 (26.9)	102 (26.4)	0.93
Solid active neoplasm (*n*, %)	16 (24.2)	90 (23.3)	0.87
≥2 comorbidities (*n*, %)	33 (49.2)	200 (51.7)	0.71
Immunomodulatory therapy (*n*, %)	20 (29.8)	104 (26.7)	0.61
Rectal swab positive for MDRO (*n*, %)	8 (11.9)	69 (17.8)	0.23

ID: infectious disease; ATD: atherothrombotic disease; CDr: exacerbation of chronic disease; MDRO: multi-drug resistant organism.

**Table 4 jcm-11-02770-t004:** Tertiary outcome. Univariate and multivariate analysis for 30-day mortality or re-hospitalization.

Variabile	Tertiary Outcome (119)	Controls (314)	Univariate Analysis	Multivariate Analysis	
			*p*	OR	CI
Male gender	66 (55.4)	174 (55.6)	0.98		
Age (year)	74.5 ± 14.7	72.7 ± 16.9	0.30		
Admission for ID	39 (32.8)	78 (24.8)	0.10		
Admission for ATD	8 (6.72)	33 (10.5)	0.23		
Admission for CDr		108 (34.4)	0.14		
Admission for other reason	22 (18.5)	94 (29.9)	0.016		
Solid active neoplasm	26 (22.0)	74 (23.6)	0.73		
≥2 comorbidities	67 (56.3)	159 (50.6)	0.29		
Immunomodulatory therapy	35 (29.4)	82 (26.1)	0.49		
Rectal swab positive for MDRO	39 (32.8)	29 (9.24)	<0.001	**4.79**	**2.79–8.23**
Development of sepsis	13 (10.9)	16 (5.1)	0.03		

OR: odd ratio; CI: confidence interval; ID: infectious disease; ATD: atherothrombotic disease; CDr: exacerbation of chronic disease; MDRO: multi-drug resistant organism.

## Data Availability

Data are available upon request to the corresponding author.

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
