# Peer review of "Clinical Implications of Multi-Drug Resistant Organisms’ Gastrointestinal Colonization in an Internal Medicine Ward: The Pandora’s Box"

_jcm, 2022, doi:10.3390/jcm11102770_

Round 1

Reviewer 1 Report

  1. Please delete double spaces throughout the manuscript
  2. Some words are merged (no space between them). Please correct it throughout the manuscript
  3. Introduction can be expanded by adding more information regarding the background of the study
  4. Line 108: Do not use bold for ‘statistical analysis’. Also, it would be better to keep the formatting suggested by the journal with the headings, the numbers, etc.
  5. Line 147: 36.1%, not 36,1%, and in line 176 also put a dot, not a comma in the number. The same is the case with the tables
  6. Line 166: ‘whereas presence of two or 166 more comorbidities was only marginally associated with the development of sepsis (66.7 167 vs 50.1%; p 0.07).’ This is not statistically significant and can be deleted. Otherwise, just say that there was a non-statistically significant trend. The same is for line 183
  7. Line 191: do you mean 6.31%?
  8. Please add the tables in the main manuscript (they are given as separate non-published material on the site, and the manuscript only contains text)
  9. Titles of the tables should be on top, and abbreviations at the footnote
  10. SD should be added to the footnotes
  11. Table 2: Sepsi? Do you mean septic? Or sepsis?
  12. The first paragraph in the methods section can be moved at the end, right before the statistical analysis. On the other hand, the definition of sepsis could be moved between line 78 and line 79 (before mentioning sepsis in the methods)

Author Response

Reviewer #1

  • Please delete double spaces throughout the manuscript

Done.

  • Some words are merged (no space between them). Please correct it throughout the manuscript

We apologize for these typing errors. We modified all the incorrectly merged words

  • Introduction can be expanded by adding more information regarding the background of the study

The introduction has been expanded as suggested (please see page 2 of the revised version of the manuscript.

  • Line 108: Do not use bold for ‘statistical analysis’. Also, it would be better to keep the formatting suggested by the journal with the headings, the numbers, etc.

 We apologize for these mistakes. The manuscript has been modified according to the journal formatting

  • Line 147: 36.1%, not 36,1%, and in line 176 also put a dot, not a comma in the number. The same is the case with the tables

  Thank you for underlying these inaccuracies. Corrections have been made

  • Line 166: ‘whereas presence of two or 166 more comorbidities was only marginally associated with the development of sepsis (66.7 167 vs 50.1%; p 0.07).’ This is not statistically significant and can be deleted. Otherwise, just say that there was a non-statistically significant trend. The same is for line 183

Thank you very much for this suggestion. This sentence has been modified accordingly

  • Line 191: do you mean 6.31%?

You are right. The text has been modified.

  • Please add the tables in the main manuscript (they are given as separate non-published material on the site, and the manuscript only contains text)
  • Titles of the tables should be on top, and abbreviations at the footnote
  • SD should be added to the footnotes
  • Table 2: Sepsi? Do you mean septic? Or sepsis?

Thank you for underlying all these inaccuracies. The manuscript has been modified accordingly.

The first paragraph in the methods section can be moved at the end, right before the statistical analysis. On the other hand, the definition of sepsis could be moved between line 78 and line 79 (before mentioning sepsis in the methods)

The text has been modified according to the reviewer’s suggestions.

Reviewer 2 Report

In the manuscript entitled “Clinical implications of multi-drug resistant organisms gastrointestinal colonization in Internal Medicine Ward: the Pandora’s box.” Para et al. present results of 14.5 months-long observational studies on gastrointestinal colonization with MDRO in the internal medicine ward and its prognostic role.

Dear Authors,

The manuscript requires major editorial editing. Below are my concerns and suggestions:

write all Latin names of bacteria in italics, eg. lines 70, 71, 75,76.

If 5 patients with negative swabs were assigned to each patient with the positive test, why the number in this group is 377?

Material and methods:

Population

Describe the method of screening for MDRO.

If the patients were screened for colonization with VRE and CPE, it can not be stated (lines 74-78) [...] that rectal swab was considered positive for MDRO in case of the presence of at least one of the ESKAPE bacteria. The swabs were not screened for S. aureus.

Was the drug susceptibility test performed for all the isolated Gram-negative rods?

How were the mechanisms of resistance (KPC, NDM, VRE, and so on) detected?

Results

Please explain why only 76% of admitted patients were evaluated with a rectal swab?

Tables and figures should be included in the main text.

Avoid using abbreviations CKD, CAD, and so on.

Line 152: what were the other bacteria?

Change names of the Tables – all are in the supplementary materials, so they can not be Table 1 and Suppl. Table 1.

Line 176: what does “pf” stand for?

Line 179: change table 4 into Table 4

Line 180: remove “.” before and […]

Discussion

Line 191: add “%” after 6.31

In the discussion results of a few studies are given, however, the results from the presented studies are not discussed deeply enough with the literature. Especially I would recommend expanding the results presented in Figure 1 and discussing them.

Also, in the discussion section, the result should be referred to the data regarding the colonization rate by various MDROs in Italy based on the data available on the ECDC webpage (Surveillance Atlas of Infectious Diseases).

 References:

The style of reference must be unified according to the publisher's requirements.

Author Response

Reviewer #2

  • Dear Authors, The manuscript requires major editorial editing. Below are my concerns and suggestions:
    • write all Latin names of bacteria in italics, eg. lines 70, 71, 75,76.

We performed an Editorial Editing as suggested.

  • If 5 patients with negative swabs were assigned to each patient with the positive test, why the number in this group is 377?

As written in the method section, we planned to enroll 5 patients with a negative swab for each patient with a positive rectal swab for MDR. During the study period, we found 77 patients with a said positive rectal swab, we planned to include about 385 patients with a negative rectal swab. After patients inclusion, we realized that 8 patients were already septic at the time of hospitalization and thus were excluded.

Material and methods:

  • Population
    • Describe the method of screening for MDRO.

First of all, we apologize for the inaccuracy and the incompleteness of this part of the Methods section and we would like to thank the reviewer for this and subsequent comments that allowed us to better clarify this important topic. Having said that, we identified bacteria through mass spectrometry  MALDI-ToF, and on every rectal swab that turned out positive for MDRO (CRE, CRAB, CRPsA, and VRE), our microbiology laboratory assessed the mechanism of resistance through real-time PCR  (please see page 3 in the Methods section of the manuscript and also see the answers to the comments below).

  • If the patients were screened for colonization with VRE and CPE, it can not be stated (lines 74-78) [...] that rectal swab was considered positive for MDRO in case of the presence of at least one of the ESKAPE bacteria. The swabs were not screened for S. aureus.

You are right and we apologize again for the inaccuracy of this paragraph. The text of the paragraph has been extensively revised to clarify that among multidrug-resistant organisms belonging to the ESKAPE group, according to the Regional Law and Guidelines, our surveillance program assesses only the presence of Carbapenem-Resistant Enterobacteriaceae (CRE), Carbapenem-Resistant Acinetobacter baumanii (CRAB), Carbapenem- Resistant Pseudomonas aeruginosa (CRPsA) and  Vancomycin-Resistant Enterococci (VRE). (please refer to page 3 in the Methods section of the manuscript and  see also the answers to the comments above and  below)

  • Was the drug susceptibility test performed for all the isolated Gram-negative rods?

The drug susceptibility test was not routinely performed for all the isolated Gram-negative rods. If the isolated organism had a remarkable clinical significance, further investigation by our microbiology laboratory was promoted. (please refer to page 3 in the Methods section of the manuscript and see also the answers to the comments above and below).

  • How were the mechanisms of resistance (KPC, NDM, VRE, and so on) detected?

 The mechanisms of resistance were assessed through PCR-real time  (please see page 3 in the Methods section of the manuscript and  see also the answers to the comments above)

Results

  • Please explain why only 76% of admitted patients were evaluated with a rectal swab?

This occurred mainly for patients’ refusal. This information has been added to the manuscript (please see results section page 5).

  • Tables and figures should be included in the main text.

Tables and figures have been included in the text as requested.

  • Avoid using abbreviations CKD, CAD, and so on.

The text has been corrected according to this suggestion.

  • Line 152: what were the other bacteria?

Pseudomonas aeruginosa or other combination of different more than one MDRO. This has been specified in the text of the manuscript.

  • Change names of the Tables – all are in the supplementary materials, so they can not be Table 1 and Suppl. Table 1.

Tables and figures have been inserted into the text. The names and numbering of the tables have been corrected accordingly.

  • Line 176: what does “pf” stand for?

This typing error has been fixed.

  • Line 179: change table 4 into Table 4

Done

  • Line 180: remove “.” before and […]

Done

Discussion

  • Line 191: add “%” after 6.31

The text has been modified according to the reviewer’s suggestion.

  • In the discussion results of a few studies are given, however, the results from the presented studies are not discussed deeply enough with the literature. Especially I would recommend expanding the results presented in Figure 1 and discussing them. Also, in the discussion section, the result should be referred to the data regarding the colonization rate by various MDROs in Italy based on the data available on the ECDC webpage (Surveillance Atlas of Infectious Diseases).

We thank the reviewer for raising this important point. The discussion has been extensively modified according to the reviewer’s suggestion.

References:

  • The style of reference must be unified according to the publisher's requirements.

Sorry for this important inaccuracy. References’ style has been unified according to the publisher’s requirements.

Reviewer 3 Report

In the article entitled "Clinical implications of multi-drug resistant organisms gastrointestinal colonization in Internal Medicine Ward: the Pandora’s box",  Para et al., examined patients admitted to Internal Medicine Unit to detect positive rectal swab for multi drug resistant organisms MDRO and to compare them with negative for MDRO patients, in order to evaluate the clinical impact of MDRO colonisation in internal medicine. Upon considering medical and demographic data of the patients, the authors suggest that the use of rectal swab at hospital admission could serve as a potential prognostic tool for MDRO patients admitted in Internal Medicine. The authors, tried to present and consider all parameters in their analyses and performed the appropriate statistical analyses. They even report the advantages and the disadvantages of their research and propose that more prospective studies are needed to confirm their preliminary results. The article seems appropriate for publication in Journal of Clinical medicine as it will be of great interest for the readers. However, the authors could enrich their data analyses with some correlation analysis and discussion on the major comorbidities, outcomes, 30 days mortality, and/or incidence of death etc. that could relate with the presence of MDRO in the patients.

Moreover, some paragraphs are very short aand can be merged with each other in order for the text to be easier to read.

Author Response

Reviewer #3

  • In the article entitled "Clinical implications of multi-drug resistant organisms gastrointestinal colonization in Internal Medicine Ward: the Pandora’s box",  Para et al., examined patients admitted to Internal Medicine Unit to detect positive rectal swab for multi drug resistant organisms MDRO and to compare them with negative for MDRO patients, in order to evaluate the clinical impact of MDRO colonization in internal medicine. Upon considering medical and demographic data of the patients, the authors suggest that the use of rectal swab at hospital admission could serve as a potential prognostic tool for MDRO patients admitted in Internal Medicine. The authors, tried to present and consider all parameters in their analyses and performed the appropriate statistical analyses. They even report the advantages and the disadvantages of their research and propose that more prospective studies are needed to confirm their preliminary results. The article seems appropriate for publication in Journal of Clinical medicine as it will be of great interest for the readers.

We thank the reviewer for his/her kind comment. We are delighted that our study has been appreciated.

  • However, the authors could enrich their data analyses with some correlation analysis and discussion on the major comorbidities, outcomes, 30 days mortality, and/or incidence of death etc. that could relate with the presence of MDRO in the patients.

We performed other supplementary analyses according to the reviewer’s suggestion and we discussed these additional end-points as suggested (please see the discussion section in the text).

  • Moreover, some paragraphs are very short and can be merged with each other in order for the text to be easier to read.

Some short paragraphs have been merged as suggested (please see the method and results section).

Round 2

Reviewer 1 Report

The manuscript has been improved now.

Author Response

Firenze, May 7th  2022

Dear Editor in Chief and Academic Editor,

please find attached the revised version of our manuscript entitled “ Clinical implications of multi-drug resistant organisms gastrointestinal colonization in Internal Medicine Ward: the Pandora’s box” (jcm-1646927).  First of all, we would like to thank the Academic Editor for his/her important comments on our paper. We added all the limitations he/she suggested and we revised results and tables for inconsistencies that have been corrected. Furthermore, the paper has been revised by an English native speaker and we hope that you and the Reviewers will find satisfactory this new version. Otherwise, we are happy to further modify and improve our study if necessary.

Ombretta Para, MD

On behalf of other authors.

Reviewer 2 Report

Dear Authors,

thank you for the improvement of the manuscript.

Author Response

(The authors gave the same response as above.)
